



# A global map of root biomass across the world's forests

Yuanyuan Huang[1,2], Phillipe Ciais[1], Maurizio Santoro[3], David Makowski[4,5], Jerome Chave[6], Dmitry Schepaschenko[7,8,9], Rose Z. Abramoff[1], Daniel S. Goll[1], Hui Yang[1], Ye Chen[10], Wei Wei[11], Shilong Piao[12,13,14]

[1]Laboratoire des Sciences du Climat et de l'Environnement, LSCE/IPSL, CEA-CNRS-UVSQ, Université Paris-Saclay, 91191 Gif-sur-Yvette, France.

[2]Commonwealth Scientific and Industrial Research Organisation, Aspendale, 3195, Victoria, Australia.

[3]Gamma Remote Sensing, 3073 Gümligen, Switzerland.

[4]INRA, AgroParisTech, University Paris-Saclay, UMR 211, F-78850 Thiverval-Grignon, France.

[5]CIRED, 45bis Avenue de la Belle Gabrielle, 94130 Nogent-sur-Marne, France.

[6]Laboratoire Evolution et Diversite Biologique UMR 5174, CNRS, Universite Paul Sabatier, 118 route de Narbonne, Toulouse, 31062 France.

[7]International Institute for Applied Systems Analysis (IIASA) Schlossplatz 1, A-2361 Laxenburg, Austria.

[8]Center of Forest Ecology and Productivity of the Russian Academy of Sciences, Moscow 117997, Russia

[9]Siberian Federal University, Krasnoyarsk, 660041, Russia

[10]Department of Mathematics and Statistics, Northern Arizona University, 86001, Flagstaff, AZ, US.

[11]State Key Laboratory of Urban and Regional Ecology, Research Center for Eco-environmental Sciences, Chinese Academy of Sciences, Beijing, 100085, China.

[12]Sino-French Institute for Earth System Science, College of Urban and Environmental Sciences, Peking University, Beijing, China

[13]Key Laboratory of Alpine Ecology and Biodiversity, Institute of Tibetan Plateau Research, Chinese Academy of Sciences, Beijing, China

[14]Center for Excellence in Tibetan Earth Science, Chinese Academy of Sciences, Beijing, China

*Correspondence to*: Yuanyuan Huang (yuanyuanhuang2011@gmail.com)





**Abstract.** As a key component of the Earth system, root plays the key role in linking Earth's lithosphere, hydrosphere, biosphere, and atmosphere. Here we combine 10307 field measurements of forest root biomass worldwide with global observations of forest structure, climatic conditions, topography, land management and soil characteristics to derive a spatially explicit global high-resolution (~ 1km) root biomass dataset, including fine and coarse roots. In total, 142 ± 25 (95% CI) Pg of live dry matter biomass is stored below-ground, representing a global average root:shoot biomass ratio of 0.25 ± 0.10. Our estimations of total root biomass in tropical, temperate and boreal forests are 44-226% smaller than earlier studies(Jackson et al., 1997; Robinson, 2007; Saugier et al., 2001). The smaller estimation is attributable to the updated forest area, spatially explicit above-ground biomass density used to predict the patterns of root biomass, new root measurements and upscaling methodology. We show specifically that the root shoot allometry is one underlying driver that leads to methodological overestimation of root biomass in previous estimations. Raw datasets and global maps generated in this study are deposited at the open access repository Figshare (https://figshare.com/articles/Supporting_data_and_code_for_A_global_map_of_root_biomass_across_the_world_s_forests/12199637)

## 1 Introduction

Roots act as a hub that connects complex feedbacks among biomes, soil, water, air, rocks and nutrients. Roots mediate nutrient and water uptake by plants, below-ground organic carbon decomposition, the flow of carbohydrates to mycorrhizae, species competition, soil stabilization and plant resistance to windfall(Warren et al., 2015). The global distribution of root biomass is related to how much photosynthates plants must invest below-ground to obtain water, nitrogen and phosphorus for sustaining photosynthesis, leaf area and growth. Root biomass and activity also control the land surface energy budget through plant transpiration(Wang et al., 2016; Warren et al., 2015). While Earth Observation data combined with field data enables the derivation of spatially explicit estimates of above-ground biomass with a spatial resolution of up to 30 meters over the whole globe(GlobalForestWatch, 2019; Santoro, 2018b), the global carbon stock and spatial details of the distribution of below-ground root biomass (fine + coarse) rely on sparse measurements and coarse extrapolation so far, therefore remaining highly uncertain.

More than twenty years ago, Jackson et al, 1996, 1997 [1,2] provided estimates of the average biomass density (weight per unit area) and vertical distribution of roots for 10 terrestrial biomes. Multiplying their average root biomass density with the area of each biome results in a global root biomass pool of 292 Pg, with forests accounting for ~68%. Saugier, et al. (2001) estimated global root biomass to be 320 Pg by multiplying biome-average root to shoot ratios (*R:S*) by shoot biomass density and the land area of each biome. Mokany, et al. (2006) argued that the use of mean *R:S* values at the biome scale is a source of error because root biomass measurements are performed at small scales, but root distributions are highly spatially heterogeneous and their size distribution spans several orders of magnitude, fine roots being particularly difficult to sample(Jackson et al., 1996; Taylor et al., 2013). With updated *R:S* and broader vegetation classes, Mokany, et al. (2006)



gave a higher global root biomass of 482 Pg. Robinson (2007) further suggested that *R:S* was underestimated by 60%, which
translated into an even higher global root biomass of 540-560 Pg. These studies provided a first order estimation of the root
biomass for different biomes, but not of its spatial details. Further, it is worth noting that estimations of the global total root
biomass have increased with time.

An alternative approach to estimate root biomass is through allometric scaling, dating back to West, Brown and
Enquist (1997, 1999)[6][7] and Enquist and Niklas (2002). The allometric scaling theory assumes that biological attributes scale
with body mass, and in the case of roots, an allometric equation verified by data takes the form of $R \propto S^{\beta}$ where *R* is the root
mass, *S* the shoot mass and $\beta$ a scaling exponent. In contrast to the studies listed above assuming the *R:S* ratio to be uniform,
this equation implies that the *R:S* ratio varies with shoot size when β is not equal to one (Cairns et al., 1997; Enquist and
Niklas, 2002; Jiang and Wang, 2017; McCarthy and Enquist, 2007; Niklas, 2005; Robinson, 2004; Zens and Webb, 2002) .
Allometric equations also predict that smaller trees generally have a larger *R:S* with $\beta < 1$ , which is well supported by
measurement of trees of different sizes (Cairns et al., 1997; Jiang and Wang, 2017; Niklas, 2005; Robinson, 2004; Zens and
Webb, 2002). The allometric equation approach was applied for various forest types, and the scaling exponent $\beta$ was
observed to differ across sites(Yunjian Luo et al., 2018), species(Cheng and Niklas, 2007), age(Cairns et al., 1997), leaf
characteristics(Luo et al., 2012), elevation(Moser et al., 2011), management status(Ledo et al., 2018), climatic conditions,
such as temperature(Reich et al., 2014), soil moisture and climatic water deficit (Ledo et al., 2018), as well as soil nutrient
content and texture(Jiang and Wang, 2017). Despite successful application of allometric equations for site- and species-
specific studies(Yunjian Luo et al., 2018), their use to predict global root biomass patterns appears to be limited and
challenging.

## 2 Methods

### 2.1 Overview

We use a new approach to upscale root biomass of trees at the global scale (Supplementary Figure 1) based on
machine learning algorithms trained by a large dataset of i*n-situ* measurements (10307) [14,30,31]of root and shoot biomass for
individual woody plants (see Methods: Field measurement, Supplementary Data), covering 465 species across 10 biomes
defined by The Nature Conservancy(Olson and Dinerstein, 2002) (Supplementary Figure 2). We compared the allometric
upscaling and tested three machine learning techniques (the random forest, the artificial neural networks and multiple
adaptive regression splines), searched through a pool of 47 predictor variables that include shoot biomass and other
vegetation, edaphic, topographic, anthropogenic and climatic variables (Supplementary Table 1) and selected the random
forest model (RF) that performs best on cross validation samples (see section Building predictive models below). Using this
RF model, we mapped the root biomass of an average tree over an area of ~1km x1 km across the globe using as predictors
gridded maps of shoot biomass (weight per area) (Santoro, 2018a), tree height(Simard et al., 2011), soil nitrogen(Shangguan
et al., 2014), pH(Shangguan et al., 2014), bulk density(Shangguan et al., 2014), clay content(Shangguan et al., 2014), sand
content(Shangguan et al., 2014), base saturation(Shangguan et al., 2014), cation exchange capacity(Shangguan et al., 2014),
water vapor pressure(Fick and Hijmans, 2017), mean annual precipitation(Fick and Hijmans, 2017), mean annual



temperature(Fick and Hijmans, 2017), aridity(Trabucco and Zomer, 2019) and water table depth(Fan et al., 2013) (Supplementary Figures 10,11,12). Combining with the tree density (number of trees per area)(Crowther et al., 2015) at the global scale, we quantified the global forest root biomass.

**2.2 Field measurements**

Our dataset was compiled from literature and existing forest biomass structure or allometry databases(Falster et al., 2015) (Ledo et al., 2018; Schepaschenko et al., 2018; Schepaschenko et al., 2017). We included studies and databases that reported georeferenced location, root biomass and shoot biomass. For example, Ref(Poorter et al., 2015) is not included due to lack of georeferenced location and Ref(Iversen et al., 2017) in not used as we also need measurements of other plant compartments like shoot biomass. Repeated entries from existing databases were removed. One of the databases(Falster et al., 2015) reported data on woody plants which also include shrub species. We kept the shrub data partly because the remote sensing products we used to generate our root map do not clearly separate trees from shrubs. Around 82% of the extracted entries also recorded plant height and management status. Height was identified as an important predictor in our model assessment, and entries were discarded when height was missing (18% of data). As woody plant age was reported in 19% of the entries only, the values of this variable was determined from another source of information, i.e. from a composite global map introduced in the next section. Species names were systematically reported, but biotic, climatic, topographic and soil information were missing for a substantial proportion of entries and values of these variables were thus extracted from independent observation-driven global maps as explained in the next section. Our final dataset includes biomass measurements collected in 494 different locations from 10307 individual plants, which cover 465 species across 10 biomes as defined by The Nature Conservancy(Olson and Dinerstein, 2002) (Supplementary Figure 2; Supplementary Data).

**2.3 Preparing predictor variables**

We used 47 predictors that broadly cover 5 categories: vegetative, edaphic, climatic, topographic and anthropogenic (Supplementary Table 1). Vegetative variables include shoot biomass, height, age, maximum rooting depth, biome class and species. Edaphic predictors cover soil bulk density, organic carbon, pH, sand content, clay content, total nitrogen, total phosphorus, Bray phosphorus, total potassium, exchangeable aluminium, cation exchange capacity, base saturation (BS), soil moisture and water table depth (WT). Climatic predictors are mean annual temperature (MAT), mean annual precipitation (MAP), the aridity index that represents the ratio between precipitation the reference evapotranspiration, solar radiation, potential evapotranspiration (PET), vapor pressure, cumulative water deficit (CWD=PET - MAP), wind speed, and mean diurnal range of temperature (BIO2 ), isothermality (BIO2/BIO7) (BIO3), temperature seasonality (BIO4), max temperature of warmest month (BIO5), min temperature of coldest month (BIO6), temperature annual range (BIO7), mean temperature of wettest quarter (BIO8), mean temperature of driest quarter (BIO9), mean temperature of warmest quarter (BIO10), mean temperature of coldest quarter (BIO11), precipitation of wettest month (BIO13), precipitation of driest month (BIO14), precipitation seasonality (BIO15), precipitation of wettest quarter (BIO16), precipitation of driest quarter (BIO17), precipitation of warmest quarter (BIO18), precipitation of coldest quarter (BIO19). The topographic variable is elevation and we take the management status (managed or not) as the anthropogenic predictor. All references are given in Supplementary



Table 1.

As *in-situ* field measurements of above-ground biomass (AGB) do not offer a full global coverage, gridded shoot biomass data were derived from satellite AGB products to predict root biomass at the global scale with a 1km by 1 km spatial resolution. The gridded global shoot biomass dataset used in our study has been extensively calibrated with in-situ observations and is currently the most reliable source of information on shoot biomass offering a global coverage (Baccini et al., 2017; Santoro, 2018a). To derive the shoot or  AGB per tree (in unit of weight per tree) to generate spatially explicit global root biomass, we combined the GlobBiomass-AGB satellite data product(Santoro, 2018a) ( in unit of weight per unit area) with a tree density map (number of trees per unit area)(Crowther et al., 2015). The GlobBiomass dataset was based on multiple remote sensing products (radar, optical, LiDAR) and a large pool of *in-situ* observations of forest variables(Santoro et al., 2015; Santoro, 2018b). The original GlobBiomass-AGB map was generated at 100 m spatial resolution; for this study, the map was averaged into a 1 km pixel by considering only those pixels that were labeled as forest (Santoro, 2018b). A pixel was labeled as forest when the canopy density was larger than 15% according to Hansen et al. (2013)'s dataset (Hansen2013) averaged at 100 m. The 1-km resolution global tree density map was constructed through upscaling 429,775 ground-based tree density measurements with a predictive regression model for forests in each biome(Crowther et al., 2015). The forest canopy height map took advantage of the Geoscience Laser Altimeter System (GLAS) aboard ICESat (Ice, Cloud, and land Elevation Satellite). Forest definitions are slightly different among these three maps. Forest area of the tree density map was based on a global consensus land cover dataset that merged four land cover products (Tuanmu and Jetz, 2014), and which gave a total tree count equal to the Hansen et al. (2013) land cover product (Crowther et al., 2015). The canopy height map used the Globcover land cover map(Hagolle et al., 2005) as reference to define forest land. We took Hansen2013 with a 15% canopy cover threshold as our base forest cover map. We approximated the missing values in tree density and height (due to mismatches in forest cover) by the mean of a 5x5 window that is centered on the corresponding pixel. We quantified the potential impact of mismatches in forest definition by looking into two different thresholds: 0% and 30%.

We merged several regional age maps to generate a global forest age map. The base age map was derived from biomass through age-biomass curve similarly as conducted in tropical regions in ref.(Poulter, 2019)  This age map does not cover the northern region beyond 35 N. We filled the missing northern region with a North American age map (Pan et al., 2011) and a second age map covering China(Zhang et al., 2017). Remaining missing pixels were further filled with the age map derived from MODIS disturbance observations. For the final step, we filled the remaining pixels with the GFAD V1.1 age map(Poulter, 2019). GFAD V1.1 has 15 age classes and 4 plant functional types (PFTs). We choose the middle value of each age class and estimated the age as the average among different PFTs.

Detailed information of all ancillary variables is listed in Supplementary Table 1. To stay coherent, we re-gridded each map to a common 1 km x 1 km grid through the nearest neighbourhood method.

**2.4 Building predictive models**

We investigated the performance of the allometric scaling and three non-parametric models: RF, ANN and MARS. Allometric upscaling relates root biomass to shoot biomass in the form of $R \propto S^{\beta}$. RF is an ensemble machine learning



method that builds a number of decision trees through training samples(Breiman, 2001). A decision tree is a flow-chart-like structure, where each internal (non-leaf) node denotes a binary test on a predictor variable, each branch represents the outcome of a test, and each leaf (or terminal) node holds a predicted target variable. With a combination of learning trees (models), RF generally increases the overall prediction performance and reduces over-fitting. ANN computes through an interconnected group of nodes, inspired by a simplification of neurons in a brain. MARS is a non-parametric regression method that builds multiple linear regression models across a range of predictors.

Tree shoot biomass from the *in-situ* observation data spans a wider range than shoot biomass per plant derived from global maps ($1 \times 10^{-7}$ to 8800 vs. $7.9 \times 10^{-5}$ to 933 kg/plant). To reduce potential mapping errors, we selected training samples with shoot biomass between $5 \times 10^{-5}$ and 1000 kg/plant. The medians and means of shoot biomass, root biomass and *R:S* from the selected training samples are similar to those from the entire database. Also, to reduce the potential impact of outliers, we analyzed samples with *R:S* falling between the 1st and 99th percentiles, which consists of 9589 samples with *R:S* ranging from 0.05 to 2.47 and a mean of 0.47 and a median of 0.36. Sample filtering slightly deteriorated model performance and had minor impact on the final global root biomass prediction (145 from whole samples vs.142 Pg from filtered data). We chose root biomass as our target variable instead of *R:S* because big and small trees contribute equally to *R:S* while big trees are relatively more important in biomass quantification. In our observation database, we have more samples being small woody plants (Supplementary Figure 6). A model with an overall good performance will not guarantee a good prediction on woody plants with higher biomass. We furthermore split the *in-situ* measured shoot biomass into three groups, namely measurements with shoot biomass smaller than 0.1, between 0.1 and 10, and larger than 10 kg/plant, and trained a specific model for each class. The rationale behind this splitting is: (1), to remove the bias of small plants from the distribution of *in-situ* measured woody shoot biomass (Supplementary Figure 6); (2), to account for the shift of root shoot allometry with tree size(Poorter et al., 2015) (Ledo et al., 2018; Zens and Webb, 2002); (3), to improve the performance of independent validation through numerous combinations of splitting trials; (4), and because tests through weighting samples or resampling samples (e.g., over-sampling using Synthetic Minority Over-sampling Technique) showed no better performance.

Model performances were assessed by 4-fold cross-validation using two criteria: the mean absolute error (MAE) and the R-squared value ($R^2$). MAE quantifies the overall error while $R^2$ estimates the proportion of variance in root biomass that is captured by the predictive model. We favoured the model with a smallest MAE, a highest $R^2$ and with minimum number of predictors. For non-parametric models, starting from a model with all 47 predictors, we sequentially excluded predictors that did not improve model performance one after another. The order of removing predictors was random. After a combination of trials, the best model was from RF and the final set of predictors included shoot biomass, height, soil nitrogen, pH, bulk density, clay content, sand content, base saturation, cation exchange capacity, vapor pressure, mean annual precipitation, mean annual temperature, aridity and water table depth.

## 2.5 Generation of the global root biomass map

Over an area of 1km x 1km, we assumed a tree with an average shoot biomass follows the RF model trained above. Building upon a large set of samples with each field measurement being an outcome of complex local interactions (including



within-vegetation competition), we implicitly accounted for some sub-pixel variability (e.g., resource competition and

responses to environmental conditions) on root biomass. We combined the RF model with global maps of selected predictor variables to produce the root biomass map which has a unit of weight per tree. This map was multiplied by tree density at 1-km resolution to obtain the final root biomass map with a unit of weight per area (Supplementary Figure 1).

**2.6 Uncertainty quantification**

We estimated the overall uncertainty of the root biomass estimates through quantifying errors caused by predicting

root biomass at the 1-km resolution ($\eta_{pred}$) and converting root biomass per tree to root biomass per unit area ( $\eta_{con}$). We quantified the prediction uncertainty through an ensemble of predictions. We collected 8 additional global predictor datasets (3 shoot biomass, 2 soil and 3 climate datasets) (Supplementary Table 2) and carried out 8x4 (4 folds) sets of additional predictions replacing the predictors by each of these additional data maps. We calculated the standard deviation among 36 predictions for each pixel (Supplementary Figure 4a). Converting root biomass from per tree to per area is through the tree

density(Crowther et al., 2015). We assumed the coefficient of variation (CV, i.e., the ratio of the standard deviation to the mean) in tree density mapping caused the same relative uncertainty in our per unit area root biomass. CV in tree density mapping at the biome scale was derived from Ref(Crowther et al., 2015) through dividing uncertainties in quantifying total tree numbers by the total tree numbers. $\eta_{con}$ in terms of standard deviation is therefore equal to the product of CV and the mean root biomass at each pixel (Supplementary Figure 4b). At last we propagated these two sources of uncertainty

assuming these errors were random and independent. Note that we did not account for uncertainties in *in-situ* root biomass measurements used in training the RF model. The overall uncertainty (standard deviation) at the pixel level was calculated through,

$$\eta_{root} = \sqrt{\eta_{pred}^2 + \eta_{con}^2} \qquad (1)$$

At the biome and global scales, we obtained total root biomass for each of the 36 predictions and estimated the standard

deviations of the total root biomass. $\eta_{con}$ was estimated by multiplying CV by biome or global-level root biomass. We propagated these two sources of uncertainty through Equation 1. Note that the semivariogram of the random forest prediction errors do not show a clear autocorrelation pattern (Supplementary Figure 10).

**2.7 Relative importance of predictor variables**

The impact of predictors on predicting *R:S* was estimated through the Spearman's rank-order correlation at both the global and biome scales. We log-transformed the *R:S* and shoot biomass before standardizing these datasets. Partial dependence plots(Hastie et al., 2009) show the marginal effect that one predictor has on root biomass from a machine learning model, and serves as a supplement to the Spearman correlation.

**3 Results**

We estimated a global total root biomass of 142±25 (95% CI) Pg (see Methods for uncertainty estimation and Supplementary Figures 3, 4) for forests when forest is defined as all areas with tree cover larger than 15% from the Hansen





et al. (2013) tree cover map. The corresponding global weighted mean *R:S* is 0.25 ± 0.10. The root biomass spatial distribution generally follows the pattern of shoot biomass, but there are significant local and regional deviations as shown by Figure 1. 51% of the global tree root biomass comes from tropical moist forest, 14% from boreal forest, 12% from

temperate broadleaf forest and 10% from woody plants in tropical and subtropical grasslands, savanna and shrublands (Supplementary Table 3). Given our use of a tree cover threshold of 15% at 30m resolution, our estimate ignores the roots of isolated woody plants present in arid or cold regions (Staver et al., 2011), as well as heterogeneous (e.g. urban or agriculture) landscapes and is possibly an under-estimate. Total root biomass decreases from 151 to 134 Pg when the canopy cover threshold used to define forest land is increased from 0% to 30%. The root biomass density per unit of forest area is highest

in tropical moist forest, followed by temperate coniferous and Mediterranean forest (Figure 1, Supplementary Table 3). Cross validation showed a good match between predictions from our RF model and *in-situ* observations (Figures 2e, all data; Supplementary Figure 7, for each biome; Supplementary Figure 8, for three tree size classes; Supplementary Figure 9, for each continent), with an overall coefficient of determination $R^2$ of 0.85 and a median *R:S* similar to validation samples (0.35 from *in-situ* observation vs. 0.38 from prediction). Root biomass of tropical, temperate and boreal forests together is 44-

226% lower compared to earlier studies (Table 1, Supplementary Table 5, see Supplementary Information Comparison with published results).

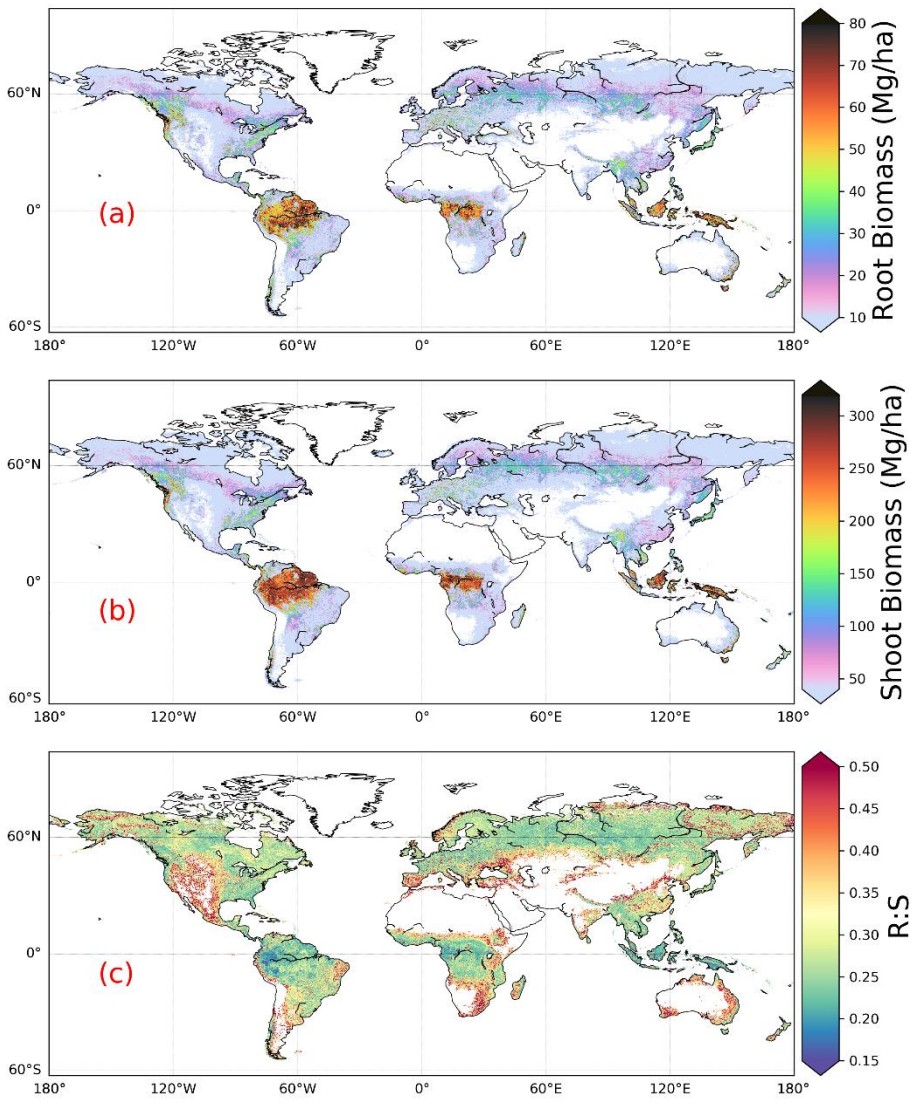

Figure 1. Global maps of forest root biomass generated through the random forest model (a), shoot biomass from
GlobBiomass-AGB(Santoro, 2018b) (b) and *R:S* (c). Forest is defined as an area with canopy cover > 15% from the Hansen
et al. (2013) tree cover map.

We then analysed the dominant factors explaining spatial variations of root biomass and *R:S* (see Methods).
Broadly speaking, locations with small trees, low precipitation, strong aridity, deep water table depth, high acidity, low bulk
density, low base saturation and low cation exchange capacity are more likely to have higher fractional root biomass (Figure
3). In line with the allometric theory, shoot biomass emerged as the most important predictor of *R:S* and root biomass, as
given by the Spearman correlation analysis shown in Figure 3, and partial importance plots (Supplementary Figures 11, 12,





13). Water related variables (precipitation, water table depth, aridity and vapor pressure) also emerged as important predictors in explaining *R:S* patterns (Figure 3)(Ledo et al., 2018), with trees and woody plants in dry regions generally having higher *R:S* (Supplementary Tables 3, 4), and with stronger dependence on precipitation especially when precipitation

is low and on water table depth when the water table is deep. Temperature is slightly negatively correlated with *R:S* at the global scale, in line with Reich et al. (2014). However, the relationship between temperature and below-ground biomass is not consistent among biomes (Figure 3) and biomass size groups (Supplementary Figures 11, 12, 13). The relationship between total soil nitrogen and root biomass is negative when soil nitrogen content is below 0.1-0.2 % (Supplementary Figure 11, 12, 13). Root biomass and *R:S* generally increases with soil alkalinity (Figure 3, Supplementary Figures 11, 12,

13). Low pH is toxic to biological activities and roots, especially as fine roots are sensitive to soil acidification, as revealed by a recent meta-analysis(Cheng Meng et al., 2019). Our results also indicate overall positive correlations between CEC, BS and R:S, but the processes that may account for these correlations are less clear from literature. Age has been shown to be important for *R:S*(Schepaschenko et al., 2018). How age regulates *R:S* remains elusive, with studies showing both positive(Waring and Powers, 2017) and slightly negative(Mokany et al., 2006) relationship between *R:S* and age. Including

forest age (see Methods: Preparing predictor variables) as a predictor only marginally improved our model prediction (see Supplementary Information for details). It is likely that shoot biomass partially accounts for age information and the quality of the global forest age data might also affect the power of this variable in improving root biomass predictions.

Table 1. Comparison between studies quantifying root biomass in tropical, temperate and boreal forests.

| | This study[S1] | This study[S2] | Jackson(Jackson et al., 1997) | Saugier(Saugier et al., 2001) | Robinson(Robinson, 2007) | This study [S3] |
|---|---|---|---|---|---|---|
| Method | Machine learning | Machine learning | Biome average root biomass density | Biome average *R:S*, shoot biomass density | Biome average *R:S*, shoot biomass density | Allometric equations |
| Tropical (Tr, Pg) | 92 | 76 | 114 | 147 | 246 | |
| Temperate(Te, Pg) | 26 | 25 | 51 | 59 | 98 | |
| Boreal (Bo, Pg) | 21 | 20 | 35 | 30 | 50 | |
| Tr + Te + Bo (Pg) | 139 | 121 | 200 | 236 | 394 | |
| Globe (Pg) | 142 | 142 | | | | 155-210 |
| RD$_{S1}$* | 0% | | 44% | 70% | 183% | |
| RD$_{S2}$& | | 0% | 65% | 95% | 226% | |

S1, Tropical moist forest (Biome 1), tropical dry forest (Biome 6), tropical/subtropical coniferous forest (Biome 11) and forest in tropical/subtropical grasslands/savannas and shrublands (Biome 3) are aggregated to represent tropical systems (Tr). Temperate broadleaf/mixed forest (Biome 4), temperate coniferous forest (Biome 5) and forest in temperate grasslands/savannas and shrublands (Biome 8) are merged together as temperate systems (Te). Boreal forest (Biome 2) and woody plants in tundra region (Biome 7) are aggregated as boreal forest (Bo). Biome classification is from The Nature





Conservancy(Olson and Dinerstein, 2002) and is shown in Supplementary Figure 2.

S2, Tropical systems (Tr): Biomes 1,6,11; Temperate systems (Te): Biomes 4,5; Boreal systems (Bo): Biome 2.

S3, Estimation based on allometric equations and the global above-ground biomass dataset from ref(Santoro, 2018b). See Supplementary Table 7 for details.

[*] RD$_{S1}$, the relative difference of Tr + Te + Bo between this study (S1) and previous quantifications. RD$_{S1}$ = (previous study –

this study)/this study x 100%. For example, in the column with the head Jackson, RD$_{S1}$ = (200-139)/139*100% = 44%.

[&] RD$_{S2}$, the same as RD$_{S1}$, but with the S2 definition of tropical, temperate and boreal systems.

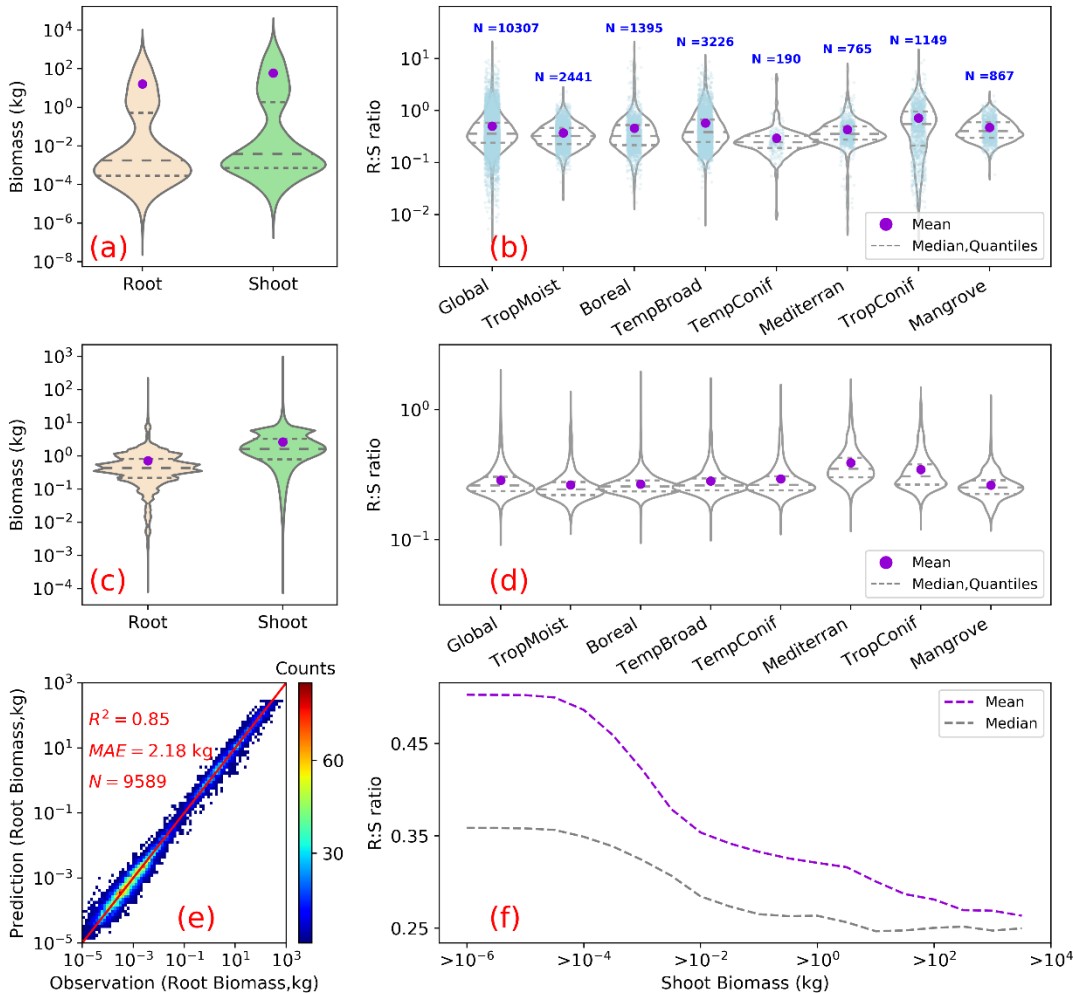

Figure 2. Root biomass and root shoot ratio (*R:S*). (a) and (b) show as violin plots the distribution of root and shoot biomass

(in unit of kg/plant) and *R:S* ratios in the raw data used for upscaling. (c) and (d) are the distributions of model-predicted





root biomass from this study, of above-ground biomass used for the prediction, and of modelled $R{:}S$ ratios at the global and biome scales. (e) is a heat plot of observed vs. predicted root biomass in kg of root per individual woody plant (see Supplementary Figures 7, 8, 9 for cross-validation at biome, tree size class and continental scales). (f) shows the mean (purple) and median (grey) $R{:}S$ as a function of shoot biomass from observations. A shift of the shoot biomass towards a

larger size ((a), (c)) results in a smaller predicted mean $R{:}S$ at the global scale ((b),(d)) (see Supplementary Table 4 for exact values) as the mean $R{:}S$ is size dependent. $R^2$ is the coefficient of determination, $MAE$ is the mean absolute error and $N$ is the number of samples. TropMoist: tropical moist forest; Boreal: boreal forest/taiga; TempBroad: temperate broadleaf and mixed forest; TempConif: temperate coniferous forest; Mediterran: Mediterranean forests, woodlands and scrub; TropConif: tropical and subtropical coniferous forest; and Mangrove forest: mangrove forest. Note that the scales of y-axis are different

between (a) and (c), (b) and (d). Model training and prediction were conducted on filtered data with $R{:}S$ falling between the 1st and 99th percentiles and shoot biomass matching the range derived from GlobBiomass-AGB(Santoro, 2018b) to reduce impacts from outliers.

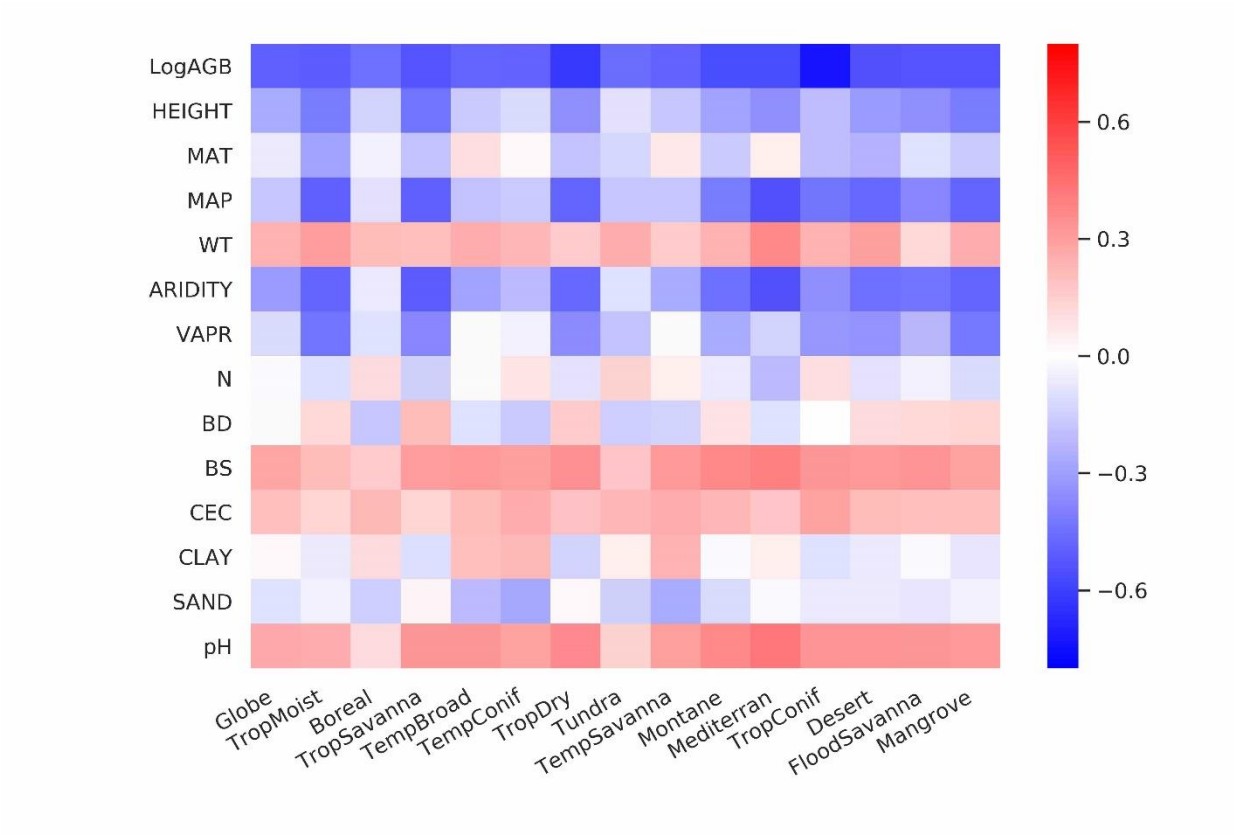




Figure 3. Spearman rank correlations between predictor variables and log-transferred *R:S*. Spearman coefficients are shown at both the global and biome scales for LogAGB: the logarithm of shoot biomass with base 10; HEIGHT: plant height; MAT: mean annual temperature; MAP: mean annual precipitation; WT: water table depth; ARIDITY: the aridity index; VAPR: water vapor pressure; N: soil nitrogen content; BD: soil bulk density; BS: soil base saturation; CEC, soil cation exchange capacity; CLAY: soil clay content; SAND: soil sand content; and pH: soil pH. From left to right, biomes are ordered by decreasing forest areas (Supplementary Figure 2).

## 4 Discussion

Our lower estimation of root biomass compared to earlier studies is attributable to differences in forest area (Supplementary Table 5), above-ground biomass density (Supplementary Table 5), root biomass measurement and upscaling methodology. For example, the forest area in temperate zones used in Jackson et al. (1997) was about one third higher than in this study. Using the root biomass density (Supplementary Table 5) and estimation method from Jackson et al. (1997), but using the updated forest area map from this study, we estimated total root biomass of tropical, temperate and boreal forests to be 147 Pg (or 184 Pg if sparse forests in tropical/subtropical/temperate grasslands/savannas and shrublands and tundra region are accounted, S1 biome definition in Table 1). This value is smaller than the 200 Pg from Jackson et al. (1997), but still larger than the 121 Pg (or 139 Pg ) (Table 1) from our machine learning approach. Our lower values of root biomass compared to Saugier et al. (2001), Mokany et al. (2006) and Robinson (2007) are caused mainly by our lower above-ground biomass density and *R:S* (Supplementary Table 5). Shoot or above-ground biomass density (AGB) of tropical zones is 70% lower in our study than in Robinson (2007) who used sparse plot data collected more than a decade ago  (Supplementary Table 5, case S2), and this lower AGB explains 27-46% of our lower root biomass (Supplementary Tables 5, 6). On the other hand, lower biome average *R:S* explains 41-48% of our underestimation compared to Robinson (2007). To elucidate this difference, we calculated weighted biome average *R:S* ratios through dividing total biome-level shoot biomass by root biomass (i.e., weighted mean *R:S*). These weighted mean *R:S,* ranging between 0.19 and 0.31 across biomes (Supplementary Table 3), are generally smaller than the *R:S* values reported in previous studies, which were based on average ratios obtained from sparser data (Supplementary Table 5). Note that the arithmetic means *R:S* (without weighting by biomass) from woody plants located in tropical, temperate and boreal zones (Supplementary Table 4) from our database are close to those from Robinson (2007) (Supplementary Table 5).

The common practice of estimating root biomass through an average *R:S* without considering the spatial variability of biomass and this ratio[4] is a source of systematic error, leading to overestimating the global root biomass for two reasons. Firstly, upscaling ratios through arithmetic averages (possibly weighted by the number of trees or area, but not accounting for the fine-grained distribution of biomass) systematically overestimates the true mean *R:S*  because *R:S* is a convex negative function of S given by $R:S \propto S^{\beta-1}$ with $\beta$ taking typical values of about 0.9 (Mokany et al., 2006; West et al., 1997, 1999) (see also Supplementary Information: Arithmetic mean *R:S* section). This explains why high-resolution S data used to diagnose weighted mean *R:S* ratios in our approach give generally smaller values than using arithmetic means across grid cells at the biome level (Weighted *R:S* Ratio in Supplementary Table 3 vs. Mean (Gridded) in Supplementary Table 4).



Multiplying this biome-level arithmetic mean *R:S* by the average biome-level shoot biomass (Supplementary Table 3) yielded a global forest root biomass of 155 Pg, larger than 142 Pg. Secondly, available measurements tend to sample more small woody plants than big trees compared to real world distributions, because small plants are easier to excavate for measuring roots (see Figure 2a, 2c) but smaller plants tend to have larger *R:S* (Figure 2e, see also Refs(Enquist and Niklas,

2002) (Zens and Webb, 2002)). This sampling bias shifts the *R:S* towards larger values. If we use the biome-level mean *R:S* from our *in-situ* database (Mean (Obs) in Supplementary Table 4), multiplying the shoot biomass (Supplementary Table 3) yielded a global value of 233 Pg, larger than using the mean *R:S* across grid cells through RF (155 Pg). Our RF approach uses *in-situ* data for training but in the upscaling, it accounts for realistic distributions of plant size (Supplementary Figure 5; Supplementary Table 4). We further verified that our upscaled *R:S* ratios are robust to sub-sampling the training data in

observed distributions, so that the bias of training data towards small plants does not translate into a bias of upscaled results (see Method, Supplementary Figure 8).

The upscaling approach using allometric equations should also tend to overestimate (see Supplementary Information: Allometric upscaling section) the global root biomass due to the curvature of these allometric functions(Enquist and Niklas, 2002; Zens and Webb, 2002). The global forest root biomass ranges between 154 – 210 Pg when root biomass

was upscaled through different allometric equations collected from literature and fitted to our database (Supplementary Table 7), generally larger than from the RF mapping. The global root biomass is likely to be smaller than when applying the allometric equation to the spatial average of shoot biomass (Supplementary Figures 14,15,16,17). Thus, future *in-situ* characterization of the distribution of tree sizes across the world's forests (see Supplementary Information: Allometric upscaling section) would greatly improve root biomass quantification. Note that how well our global estimate reflects the

real root biomass is conditioned upon the accuracy of *in-situ* root measurement database used to train our RF model. Under-sampling is a common issue in many root studies due the fractal distribution of root systems in soils and the difficulty of implementing an efficient sampling strategy(Taylor et al., 2013), especially for large trees. We did not quantify the uncertainty of our estimates associated with *in-situ* root measurements due to lack of reliable information.

An accurate spatially explicit global map of root biomass helps to improve our understanding of the Earth system

dynamics by facilitating fundamental studies on resource allocation, carbon storage, plant water uptake, nutrient acquisition and other aspects of biogeochemical cycles. For example, the close correlation (correlation coefficient: 0.8) between root biomass and rooting depth(Fan et al., 2017) at the global scale and the importance of roots for plant water uptake and transpiration reflect close interactions between vegetation and hydrological cycles. The quest for drivers that affect allocation and consumption of photosynthetic production is a major focus of comparative plant ecology and evolution, as well as the

basis of plant life history, ecological dynamics and global changes(McCarthy and Enquist, 2007). Turnover time and allocation are two key aspects that contribute to large uncertainties in current terrestrial biosphere model predictions(Bloom et al., 2016; Friend et al., 2014). Our root biomass map does not provide data on turnover or allocation, but an outcome on their aggregated effects. Future studies combining the root biomass map with upscaled root turnover data could shed light on the allocation puzzle. The growth of the fast turnover part of roots, mostly fine roots, and leaves are highly linked. If we



assume an annual turnover of leaves and fine roots, a preliminary estimation of average forest fine root biomass (from leaf biomass) reaches 6.7-7.7 Pg (see Supplementary Information: Preliminary estimation of fine root biomass). Despite being a small portion of total plant biomass and highly uncertain, fine roots are temporally variable and functionally critical in ecosystem dynamics. Future studies on global distribution and temporal dynamics of fine roots are valuable. Considering specific biomes, tropical savannas would benefit from better root biomass estimation due to its large land area, and in

tropical dry forests, field measurements of root and shoot biomass are needed to refine root biomass quantifications.

**Data availability**

Raw datasets and global maps generated in this study are deposited at the open access repository Figshare (https://figshare.com/articles/Supporting_data_and_code_for_A_global_map_of_root_biomass_across_the_world_s_forests/ 12199637) (Huang et al., 2020). The source data and code underlying Figs 1, 2, 3 Supplementary Figs 2-17 are also provided

at the Figshare.

**Code availability**

Calculations were conducted through Python 2.7.15 and ferret 6.72. The code is deposited at the open access repository Figshare

(https://figshare.com/articles/Supporting_data_and_code_for_A_global_map_of_root_biomass_across_the_world_s_forests/

12199637) (Huang et al., 2020).

**Author contributions**

Y.H. and P.C. designed this study. Y. H., P.C., M.S., J.C and D.S. collected the data. D.M., P.C., M.S., J.C., Y.C. and Y.H. discussed analyzing methods. Y.H. conducted the analysis and drafted the manuscript. All authors discussed the results and contributed to the manuscript.

**Competing interests**

The authors declare that they have no conflict of interest.

**Acknowledgements**

Y.H., D.S.G and P.C. received support from the European Research Council Synergy project SyG-2013-610028 IMBALANCE-P and P.C. and Y.H. from the ANR CLAND Convergence Institute. H.Y., D.S. and M. S. were funded

through the ESA Climate Change Initiative BIOMASS project. Collecting Russian data were supported by The Russian Science Foundation (project no. 19-77-30015). R.Z.A. received support from the French government grant "Make Our Planet Great Again".

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
