# Peer review of "A global map of root biomass across the world's forests"

_Earth System Science Data, 2021_

## Author Comment (AC1)

The manuscript would benefit if the authors could provide the reference depth their root biomass estimates are referring to. Is it top 100 cm, 30 cm, or all roots until bedrock? What is the sampling depth of the individual studies? Can we be sure that all the roots were sampled or would it make sense to include sampling depth as an additional predictor in the random forest?

**Response: Thank you for the constructive comments. In our process of building the machine learning models, we did consider the depth related variables (section 2.3). We tested the maximum rooting depth as a candidate predictor (Line 124, tracked version of the manuscript). However, the maximum rooting depth had a minor impact on the performance of the machine learning model and was not selected as a final predictor. The maximum rooting depth was extracted from a global database as most entries in our observation databases did not explicitly report to which depth people extracted roots. In root studies, researchers generally dig deep into the soil until they could extract most of the roots. In our database, observations likely vary in depth. We do not have a standardize reference depth. Different species growing under different environments are likely have different rooting depth. We assumed individual root studies (which reported extracting most of the root) are valid and predicted total root biomass. In our current database, we have very limited information on the extracting depth from each individual study. The limited impact of maximum rooting depth in our test remain to be confirmed as the uncertainties in maximum rooting depth may also play a role. Future root measurements with standard protocol and detailed depth information are likely to be helpful in improving large scale root studies. We added "Note the maximum rooting depth had a minor impact on model performance and was not selected in the final model. The depth to which roots inhabit varies among species and environment. Our model predictions are therefore not specific to a certain soil depth." around Line 215 to make this point clearer.**

---

## Author Comment (AC2)

General comments

This manuscript describes a new high resolution, global map of root biomass—crucially, one generated by a different approach than most previous studies, that produces a very different (lower) result globally. This is an interesting and important topic, and the resulting dataset will be a valuable resource for a wide range of scientists. The text is generally clear and well written, methods clearly described, and uncertainty and cross validation quantification comprehensive. I have not tried to rerun the code posted on Figshare but applaud its availability, which is crucial for scientific transparency and reproducibility. From scanning through, it looks clear and complete.

**Response: Thank you.**

There are a few problems. I echo the previous reviewer's point about sampling depths—this should be more clearly described. There also is a key reference and comparison dataset that isn't cited but almost certainly should be: Spawn, S. A., Sullivan, C. C., Lark, T. J., and Gibbs, H. K.: Harmonized global maps of above and belowground biomass carbon density in the year 2010, Sci Data, 7, 112, 2020. http://dx.doi.org/10.1038/s41597-020-0444-4. Finally, the text has a few unclear or awkward points (see short list below).

In summary, this is a strong and interesting manuscript documenting a valuable global dataset, and I think it will generate much interest in understanding the discrepancies between these and previous results. It needs minor to moderate revisions for clarity and to include the recent Spawn et al. paper and dataset.

**Response: Thank you. Please see our responses to previous reviewer's point about the sampling depth. Thank you for referring to this nice recent study. We added the study of Spawn et al 2020 to the abstract, Table 1, and the discussion. Please check below our point-by-point responses to your comments.**

Specific comments

Line 31: perhaps "a key role"

**Response: We modified the text as suggested.**

37: here and elsewhere, I'm puzzled the exclusion of recent Spawn et al. 2020

**Response: We added the Spawn et al. 2020 to Table 1, abstract and the discussion. In the discussion, we stated "Spawn et al. (2020) estimated root biomass from shoot biomass and the correlation between root/total biomass and temperature. Among previous studies, the recent study from Spawn et al. (2020) shows the smallest difference from our study. After accounting for sparse forests, the 32% smaller estimation from our study is likely linked to differences in the definition of forests and upscaling methodology" (around Line 365 of the tracked version manuscript).**

65-66: I agree this (increasing with time) is interesting, but you never return to this point in the discussion...why do you think this occurred?

**Response: We added "likely associated with improved methods in excavating roots that reduce under-sampling" as a possible reason for this increasing with time**

180-181: this sentence is awkward and either out of place, or not well connected to the material around it. Rework, probably starting a new paragraph for readability

**Response: We deleted "A model with an overall good performance will not guarantee a good prediction on woody plants with higher biomass" to make this part more readable.**

188-: the cross-validation step is crucial and I feel like this is a little light on the details. For example, the continental cross-validation is only described in the Figure 9 caption I think; should be here as well.

**Response: In the main texts around Line 270 (tracked manuscript), we explained cross validation and the model performance. We added a supplementary Table 8 to compare the performance of the random forest, the allometric fitting and another two machine learning algorithms. We also added "By continents, the performance of RF is worst in Africa (R2 = 0.6; MAE = 44 kg) partly due to limited observations (Supplementary Figure 9)". Through these changes, we improved the coverage of the cross-validation.**

262: probably start new paragraph here

**Response: We started a new paragraph as suggested.**

Table 1: see #2 above

**Response: We added Spawn et al. 2020 to Table 1.**

333-: interesting!

**Response: Thank you**

347-348: this is a good and succinct point; include in abstract?

**Response: Thanks a lot for this nice suggestion. We use this example to show, if we use the biome-level mean R:S from observations to estimate root biomass, we would overestimate the total root biomass. Because in observations, sampling is biased towards small trees. It is one of the several reasons that non-spatially explicit method might overestimate the root biomass. After a second thought, we decided not to put it into the abstract to reduce the risk of mis-interpretation of the number 233 Pg, as it requires long text to put the context.**

Supplementary l. 96-97: this is crucial detail and should be included in the F7 and F8 captions as well.

**Response: Thank you. We added this information to the captions of F7 and F8**

---

## Author Comment (AC3)

Like the previous reviewer, I commend the authors for a well-written and comprehensive manuscript describing a new, impressive and important dataset representing the biomass stocks embodied in global tree roots. I particularly appreciated the thorough retrospective in the introduction describing the history of root biomass estimation at the global scale, the simple ecological analyses the authors include in their discussion, and the transparency with which they describe [most of] their methods. Combined, I think these aspects make the manuscript an accessible descriptor that will appeal to readers across many disciplines.

**Response: Thank you.**

My only substantial concern pertains to the way in which the author's 'central' estimate was determined distinct from their uncertainty estimate. The 'central' estimate (142 Pg) was generated using models trained and executed using covariates each represented by a singular data source. The corresponding uncertainty estimate, though, was generated using an ensemble approach that used multiple [alternated] data sources to represent those covariate values. It's unclear to me why the authors favored the data sources they did when generating their 'central' estimate and, having gone the lengths of creating an ensemble, why they didn't instead just use the ensemble mean or median as the 'central estimate? I ask that at the very least, the underlying rationale be made clear in the text.

**Response: Thank you for the constructive suggestions. Aboveground biomass is the most important predictor of the root biomass. We decided not to use the average among multiple aboveground biomass products as these products vary in quality. We think the recent GlobBiomass dataset is more reliable than other cited global datasets. Spawn et al., 2020 also acknowledged the fidelity of GlobBiomass and explained the rationale to use it in their study. For environmental datasets, for some variables we could get access to multiple sources, while for some variables, we have only one data source. For consistency, if one database could provide us several variables needed in our study, we favoured this database as inconsistency among databases (e.g., spatial and temporal resolutions) also brings errors in the estimation. We added the justification for why we used the "central estimate" around Line 140. Here and after, Line numbers in responses refer to the tracked manuscript.**

In addition, I have several minor suggestions (largely related to clarity) that I ask the authors to consider and address:

Line 31: change "root plays the" to "roots play a"

**Response: We modified the text as suggested.**

Lines 55-80: This is a really nice summary of the discipline to date.

**Response: Thank you.**

Line 85: please add "n = " before 10307.

**Response: We added "n="**

Lines 90-91: Consider rephrasing to emphasize that *after comparing the results of all three of the candidate techniques*, you chose the RF approach because it performed best and only used it (not the others) for subsequent mapping and analysis. Right the text seems to abruptly drop any reference to the other two approaches (ANN and MARS).

**Response: We modified the text as suggested.**

Line 98: "Combining ______ with tree density..." please fill in the blank I've added to the text.

**Response: We changed the text into "Combining our map of root biomass per tree with tree density".**

Lines 103-105: Could these data be available from the authors? Was an attempt made to find out?

**Response: This dataset was deposited at the figshare. We added this note.**

Section 2.3: Text explaining why you chose the covariates you chose and why you selected particular datasets (over others) to represent those covariates is needed in this section.

**Response: We chose these variables based on their relevance to root dynamics from field studies and their availability at the global scale. We added the explanation around Lines 123-124 (tracked manuscript).**

Lines 124-129: Specifying that the "BIO_" variables are simply the WorldClim bioclimatic indicators help clarify why these [otherwise] seemingly odd abbreviations are used.

**Response: We added "BIO is the abbreviation for WorldClim bioclimatic indicators" to the main text.**

Line 129: Why not also include slope and/or aspect?

**Response: We did not have the slope and aspect as topographic factors after a preliminary test which shows minor effect of elevation from our database. We assume slope and aspect would also have minor effects for our global dataset as samples from mountainous and non-mountainous regions do not show big differences. It is likely elevation (and also slope and aspect) indirectly affect our prediction through modifying climate, vegetation and soil factors. We therefore did not include slope and aspect. We added this explanation at Lines 136-137.**

Line 130: Personally, I think it would be useful for table S1 to be included in the main text.

**Response: We understand Reviewer's preference. We feel Table S1 is long and makes the main text less readable. We therefore opted to keep this Table as supplementary information.**

Line 134-136: This sentence is unclear to me. Are you saying that the two input datasets by Baccini and Santoro are the most reliable sources or that the layer you derive from them is? Please clarify in the text.

**Response: We deleted the reference to Baccini to reduce confusion. Baccini focused on the tropical region. Our study targeted at the global scale and we did not merge different regional products. Santoro is therefore the right reference.**

Line 138: Did you also use the Baccini dataset? From the preceding text and references I was led to believe that you somehow combined the GlobBiomass and Baccini maps but here it seems like you only use GlobBiomass. Whichever is true, please clarify in the text.

**Response: We only used GlobBiomass. As above, we deleted the reference to Baccini.**

Line 145: Please include an explicit reference to the canopy height map, here.

**Response: We added the reference Simard et al., 2011.**

Line 148: Please clarify: I don't believe Hansen reports a tree count, just area. So how can the consensus dataset give the same tree count as Hansen?

**Response: Sorry for the confusion. Crowther et al., 2015 tested on both the Tuanmu and Jetz (2014) and the Hansen et al. (2013) land cover maps, which give the same tree count. We modified the text as "Crowther et al. (2015) showed the total tree count from tree density map based on the Tuanmu and Jetz (2014) land cover is the same as from the Hansen et al. (2013) land cover product" (Lines 160-161)**

Line 152: Given your amendments and modifications described in this paragraph, can you state explicitly here what definition of 'forest area' your map adopts? This'll be important to facilitate future comparisons much like the comparison you make here.

**Response: In our study, we define the forest based on the tree canopy coverage. "A pixel was labeled as forest when the tree canopy density was larger than 15% according to Hansen et al. (2013)'s dataset (hereafter Hansen2013) averaged at 100 m" (Lines 152-153). We also tested the impact of forest definition through varying the threshold value from 15% to 0% and 30%.**

Lines 153-159: Do all of these age maps use the same reference year? I.e. Age in/as-of what year? Please clarify in the text.

**Response: Not all age maps use the same refence year. GFAD V1.1 represents the 2000-2010 era, where the Zhang map was derived for the period 2009-2013. We added the refence year information around Lines 170-175. We used these maps as a coarse approximation of age information and mentioned "the quality of the global forest age data might also affect the power of this variable in improving root biomass predictions." (Line 301).**

Line 163: Does this imply that one candidate map was made by simply applying allometric equations to a map of forest height (and presumably stratified by taxa)? If so, that exercise isn't yet clearly explained in the methods text, here. Please do so.

**Response: We compared among the allometric equation and three machine learning algorithms. For the allometric equation, we assume one universal equation is underlying the nature of aboveground and belowground biomass partitioning, as reported in Enquist and Niklas (2002). We did not stratify our data by taxa. We added this explanation to Lines 182-183. Supplementary Table 7 also provided information on the allometric fitting.**

Line 163: Please also provide the long-form names when introducing these acronyms.

**Response: We provided the long-form names as suggested.**

Line 190: It's hard to know how you actually chose your final model given that you considered three distinct criteria. Can you elaborate here a little bit? Presumably you considered a hierarchy in these criteria? For example, if one model had the highest MAE and another had the highest R2, which did you choose? Why? How?

**Response: We considered MAE, R2 and the number of predictors. We chose the model that has the lowest MAE, highest R2. If models have comparable R2 and MAE, we chose the model with smallest number of predictors. In our tests, these criteria allowed us to choose a reasonable model. Please check on Supplementary Table 8 we added in this revision, on the performance of different models. The model has a high R2 generally also has a low MAE. If R2, MAE and the number of predictors did not give us a unique choice, bringing in another metric would be an option. In our case, these three criteria works. Please also check our response to the following comment.**

Line 190: From an ecology perspective, I can understand that minimizing the number of covariates can help more clearly explain the drivers of predicted patterns (e.g. the variable importance assessment described below). But, from a mapping perspective, where you ultimately rely on RF and where I would think accuracy is the ultimate goal, I wonder if it would be more appropriate to retain all predictors? Can you at least explain in the text your reasoning for culling covariates/favoring parsimony?

**Response: In our case, when we put all variables in our model building, we actually got a lower R2 or a higher MAE. Our previous explanation is likely to cause confusion. We modified the text as ""We favoured the model with a smallest MAE and a highest R2. For models with comparable MAE and R2, we favoured the model with the minimum number of predictors" (Line 210). We gave the priority to R2 and MAE, and then the number of predictors.**

Line 193: Consider including a table with the validation stats of each approach to clearly illustrate why RF was chosen over the others.

**Response: We added Table 8 for the performance of different machine learning methods.**

Lines 200-201: This sentence is vague. I assume you mean you combined it with the Crowther map? Either way, please make this step clearer in the text.

**Response: Yes. The random forest model generated predictions of root biomass per individual tree because our training data is based on individual trees. So we used the tree density (how many trees per area) map to estimate the root biomass per area. To make it clearer, we rewrote this part as "Our RF model was built upon individual woody trees. We combined the RF model with global maps of selected predictor variables to produce the map of root biomass for an average woody tree which has a unit of weight per tree. This map was multiplied by tree density (number of woody trees per area) (Crowther et al., 2015) at 1-km resolution to obtain the final root biomass map with a unit of weight per area (Supplementary Figure 1). " (Around Line 225)**

Section 2.6: Above you used a single dataset for each covariate to generate your 'central' estimate of root biomass. Here you describe using a separate ensemble approach to generate your error estimates. Why not instead use the mean/median of the ensemble approach as your 'central estimate'? In other words, why did you decide to prioritize the datasets listed in table S1 over those in S2? At the very least, this should be clearly explained in the text.

**Response: Please see our responses above.**

Line 230: I suggest noting that this is Pg of dry biomass so that it's not wrongly compared to estimates by others that might be in units Pg C.

**Response: We added a note saying "Note here we reported values in unit of Pg of dry biomass instead of Pg C"**

Lines 231-232: Is this the actual definition of forest cover you used? Above in the methods, it seemed a little more convoluted than this?

**Response: Please see our responses above.**

Discussion/Figure 1: Both Mokany et al. 2006 and the 2019 refinement to the IPCC guidelines report mean/median R:S's that are larger than 0.50 for some ecotypes --namely some dry woodlands, savannas, oak forests, and boreal forests. Visually it looks like your predicted patterns generally agree with these sources (at least in a relative sense) but because the colour scale is capped at 0.50,

it's hard to know if the magnitude of the estimates in these areas are comparable? Are they? Perhaps an R:S comparison with one or both of these widely used sources would be useful to include in the discussion?

**Response: Thanks for this nice point. Our predictions of locations with high R:S values are in line with Mokany et al. 2006 and the 2019 refinement to the IPCC guidelines report. We added to the discussion "And our predicted pattern of relatively large R:S in regions such as dry forests, savannas and boreal tundra woodlands (Figure 1c) are in line with data compiled from Mokany et al. (2006) and the 2019 IPCC refinement (IPCC2019, 2019)" (Around Line 365 )**

Table 1: The new maps by Spawn et al. (2020) may also be worthy of comparison here in Table 1. Near the end of their manuscript, they report 122 Pg C in global root biomass with 28.3 Pg C embodied in herbaceous (i.e. grass) roots. So the comparable value is likely 94 Pg C in tree biomass which, if you simply assume wood biomass contains 50% C, equates to 188 Pg biomass. Note too that they appear to use a more liberal definition of tree cover than you.

**Response: We added the result of Spawn et al. (2020) to Table 1. We also added this point to the abstract and discussion (Around Line 365 ).**

Lines 326-330: A similar comparison with the 2019 IPCC refinement and or Mokany would likely be well-cited but is not required.

**Response: Please see our responses above. We added this point to our discussion.**

Lines 333-363: Nice point and supporting analysis.

**Response: Thank you.**

**Reference**

**Enquist, B.J., Niklas, K.J. (2002) Global allocation rules for patterns of biomass partitioning in seed plants. Science 295, 1517-1520.**

---

## Author Response (AR2)

I thank the authors for their thorough revisions. I feel they have further improved an already excellent manuscript. I have just a few remaining concerns that I ask the authors to consider and address.

**Response: Thank you again for helping improve our study. Please see below our responses.**

Lines 88-89 – To make it clearer that the allometric results are compared to those of the machine learning approaches, please change this sentence to read "We compared the results of allometric upscaling to those of three machine learning techniques…".

**Response: Thank you. We clarified this part as suggested.**

Lines 136-137 – I wouldn't necessarily expect slope and/or aspect to covary with elevation. A point at a given elevation could feasibly have any slope and be facing any direction. While, it is acceptable that you didn't include these to variables as covariates (we can never include everything!), the justification you give is not logically sound. Please consider revising this explanation.

**Response: Thank you. We removed our explanation about the slope and aspect in the main text.**

Lines 140-141 – I'm still not satisfied by your justification for prioritizing one dataset over another when representing a given suite of covariates. More information is needed in the text. You say that you "favoured the database that is known to have higher quality data…". How did you determine the relative quality of comparable datasets? For example, Is the Shangguan et al. soil dataset better than the new SoilGrids 2.0 dataset (https://soil.copernicus.org/preprints/soil-2020-65/soil-2020-65.pdf). If so, based on what standard? Both datasets appear to contain maps of all the same soil properties.

**Response: We did not use SoilGrids 2.0 partly because this dataset does not have all the soil variables we need. SoilGrids 2.0 has bulk density, cation exchange capacity, coarse fragments, nitrogen, pH, organic carbon content, soil texture fractions. Shangguan et al. provides data for more soil variables. In addition to variables listed from SoilGrids 2.0, Shangguan et al. also has, for example, soil phosphorus, soil potassium, base saturation etc. We chose Shangguan et al. as it could allow us to test on more variables in a consistent way. In the final model, we have base saturation as one of the final predictors. SoilGrids 2.0 does not provide this variable. We do not have solid information on which dataset has better quality. Shangguan et al. dataset could provide us a more comprehensive set of soil variables in a consistent way. In the revision, we updated the format of some references to follow the journal style. We changed the reference of Shangguan et al. to Wang et al., 2014 to correct a mistake on first vs. last name. For the reason to choose Shangguan et al. dataset vs. SoilGrids 2.0. We added "For example, we used the GSES soil database (Wang et al., 2014) for the "central" estimate instead of SoilGrids 2.0 (Poggio et al., 2021) as the latter does not have the whole set of soil property variables needed in this study."**

Line 146 – Santoro et al now have a peer reviewed manuscript describing their product that is likely worth citing: https://essd.copernicus.org/preprints/essd-2020-148/

**Response: We added the reference**

**Santoro, M., Cartus, O., Carvalhais, N., Rozendaal, D., Avitabilie, V., Araza, A., de Bruin, S., Herold, M., Quegan, S., Rodríguez Veiga, P., Balzter, H., Carreiras, J., Schepaschenko, D., Korets, M., Shimada, M., Itoh, T., Moreno Martínez, Á., Cavlovic, J., Cazzolla Gatti, R., da Conceição Bispo, P., Dewnath, N., Labrière, N., Liang, J., Lindsell, J., Mitchard, E. T. A., Morel, A., Pacheco Pascagaza, A. M., Ryan, C. M., Slik, F., Vaglio Laurin, G., Verbeeck, H., Wijaya, A., and Willcock, S.: The global**

forest above-ground biomass pool for 2010 estimated from high-resolution satellite observations, Earth System Science Data Discussion, https://doi.org/10.5194/essd-2020-148, 2020.

Line 265 – Here you note a range of estimates based on the canopy cover threshold used. Since you have this data, it would seem more appropriate to compare the Spawn et al. estimate (188 Pg) to your 151 Pg (corresponding to a 0% canopy cover threshold) since Spawn et al. describe their estimate as encompassing all trees (i.e. they effectively use a 0% threshold). Thus comparing to 151 Pg rather than 142 Pg would yield a more direct comparison and allow you to say with greater confidence that the remaining difference is strictly due to differences in upscaling methodologies. By not making a direct comparison, you give the impression that difference is greater than it may be actually, which could be misleading. Note that this issue effects comparisons made in Table 1 and the abstract and related text in the discussion section

**Response: We modified the comparison with Spawn et al. (188 Pg). Earlier we compared with 142 Pg from our study with a 15% tree cover threshold for forest definition, and we pointed to forest area and upscaling methodology as potential reasons for the differences. In this revision, we compare between Spawn et al. and our estimate with the 0% tree cover threshold for forest definition. We point out the difference is mostly likely due to the upscaling methodology, but we could not exclude out the slight differences in forest (woody tree) definition as Spawn et al. used modified aboveground biomass for Africa and tundra. The revised text read as "Compared to our estimation with the 0% tree cover threshold for forest definition (i.e., 151 Pg root biomass), the 24.5% higher estimation from Spawn et al. (2020) is most likely linked to the upscaling methodology, in addition to the slight difference in the definition of forest (woody) area especially in Africa and Tundra.". We also updated values in abstract and Table 1.**

Line 367 – What do you mean by "After accounting for sparse forests"? Please make this more clear in the text.

**Response:  We rewrote this part without reference to sparse forests. Please see our response above.**

Line 367 – Based on the description of this calculation given in the footnotes of table 1, the 32% estimate given here is calculated as a percent *increase*. However, here, you present it as though its a percent *decrease* by saying "32% smaller" which is inaccurate. To illustrate: 188*(1-0.32) = 128 not 142. Conversely 142*(1+0.32) = 188. In reality, the 188 Pg estimate is 32% *greater* than your 142 Pg estimate or your 142 Pg estimate 24.5% *less* than the 188 Pg estimate. (Further, if you instead compare to the 151 Pg estimate as I suggest above, your estimate (151 Pg) is 19.7% less than 188 Pg and 188 Pg is 24.5 greater than your estimate). Please make sure the language surrounding this comparison is accurate both here and in the abstract.

**Response: Thanks a lot for pointing out this twist. We updated the language in abstract and the main text. Now we use "larger" to shift the base to estimations of our study.**